# Ultra-narrow-band near-infrared thermal exciton radiation in intrinsic one-dimensional semiconductors

Taishi Nishihara [1,2], Akira Takakura[1,2], Yuhei Miyauchi [1,2,3] & Kenichiro Itami [1,2,4]

Thermal radiation is the most primitive light emission phenomenon of materials. Broadband radiation from red-hot materials is well known as the kick-starter phenomenon of modern quantum physics in the early twentieth century; even nowadays, its artificial control plays a central role in modern science and technology. Herein, we report the fundamental thermal radiation properties of intrinsic one-dimensional semiconductors and metals, which have not been elucidated because of significant technical challenges. We observed narrow-band near-infrared radiation from semiconducting single-walled carbon nanotubes at 1000–2000 K in contrast to its broadband metallic counterpart. We confirm that the ultra-narrow-band radiation is enabled by the thermal generation of excitons that are hydrogen-like neutral exotic atoms comprising mutually bound electrons and holes. Our findings uncover the robust quantum correlations in intrinsic one-dimensional semiconductors even at 2000 K; additionally, the findings provide an opportunity for excitonic optothermal engineering toward the realization of efficient thermophotovoltaic energy harvesting.

---

[1] JST-ERATO, Itami Molecular Nanocarbon Project, Nagoya University, ChikusaNagoya 464-8602, Japan. [2] Graduate School of Science, Nagoya University, ChikusaNagoya 464-8602, Japan. [3] Institute of Advanced Energy, Kyoto University, UjiKyoto 611-0011, Japan. [4] Institute of Transformative Bio-Molecules (WPI-ITbM), Nagoya University, ChikusaNagoya 464-8602, Japan. Correspondence and requests for materials should be addressed to Y.M. (email: miyauchi@iae.kyoto-u.ac.jp) or to K.I. (email: itami@chem.nagoya-u.ac.jp)

Thermal radiation is the most primitive form of light emission from materials; hot materials emit photons through radiative relaxation of thermally excited particles (e.g., electrons) to lower energy levels. In contrast to the narrow-band radiation that is observed from isolated atoms because of their well-defined energy levels, condensed matter generally emits a broad spectrum of radiation owing to its continuous energy bands (Fig. 1a, b). This broadband thermal radiation is well known as the kick-starter phenomenon of modern quantum physics in the early twentieth century[1]; even nowadays, its artificial control plays a central role in many aspects of modern science and technology, from thermal energy harvesting[2–7] to the aerospace industry[8]. Because of the subject's long history, the physics of thermal radiation phenomena may appear to be completely understood. With regard to one-dimensional (1D) systems, however, the fundamental quantum aspects on their thermal radiation phenomena are still unclear.

One of the most intriguing aspects of 1D solids is that they have characteristics of both atom-like discreteness and solid-like continuity in their energy spectra (Fig. 1c). Moreover, correlations among electrons and/or holes become significant in 1D solids because of unavoidable spatial overlap of the quantum wavefunctions in 1D space, which substantially modifies the nature of the excited states. Several 1D optical properties were reported in semiconducting quantum wires, molecular wires, and carbon nanotubes[9]. However, the fundamental high-temperature radiative properties of intrinsic 1D systems remain unclear mainly because of significant technical challenges related to preparing real 1D materials with ultra-high-temperature stability, precise temperature measurement of the system, and heating the materials while maintaining their charge neutrality.

The most promising candidate materials for exploring 1D optothermal science are single-walled carbon nanotubes (hereafter referred to as nanotubes; inset of Fig. 1d). These are ultimately thin quasi-1D systems with outstanding thermal stability even at 2000 K[10], and they may be semiconductors or metals depending on their sidewall structure[11–13]. As predicted for an ideal 1D semiconductor[14], it is illustrated that almost the entire oscillator strength is concentrated in the exciton states, which are hydrogen-like bound state of electrons and holes, of the semiconducting nanotubes[15–17], even at room temperature, because of the outstanding electron correlation resulting from strong 1D quantum confinement and weak dielectric screening (Fig. 1d). These unique aspects of nanotubes should inspire exploration of the fundamental high-temperature optothermal properties of 1D solids. Indeed, some pioneering studies reported high-temperature radiation from carbon nanotubes under Joule-heating conditions[18–21], exhibiting light emission spectra consisting of broad peaks (full-width at half-maximum (FWHM) of ~350–400 meV)[19,20] or no peak features at all[18,21]. However, the carrier doping and current injection required to heat the nanotubes modify their 1D quantum correlation effects[22], and the origin of the peak features (whether they are band-to-band or excitonic transitions) remains debatable[19,20]. In addition, the possibility of competing electroluminescence mechanisms, including ambipolar carrier injection and impact excitation[18–21,23], further complicates the interpretation of light emission phenomena during current injection.

Here, we report the fundamental thermal radiation properties of intrinsic semiconducting and metallic nanotubes suspended in vacuum. At 1000–2000 K, an intrinsic semiconducting nanotube emitted linearly polarized, narrow-band near-infrared radiation, in contrast to its broadband 1D metallic counterpart. We confirm that the narrow-band radiation originated from radiative recombination of thermally generated excitons. The exciton correlation concentrates the oscillator strength into discrete energy levels, which gives rise to the ultra-narrow-band thermal radiation. Our findings uncover the fundamental high-temperature photophysics of 1D solids, which are dominated by robust quantum correlations even at 2000 K, and may also provide an opportunity for excitonic optothermal engineering that enables the conversion of heat to photons with a well-defined energy toward the realization of efficient thermophotovoltaic energy harvesting.

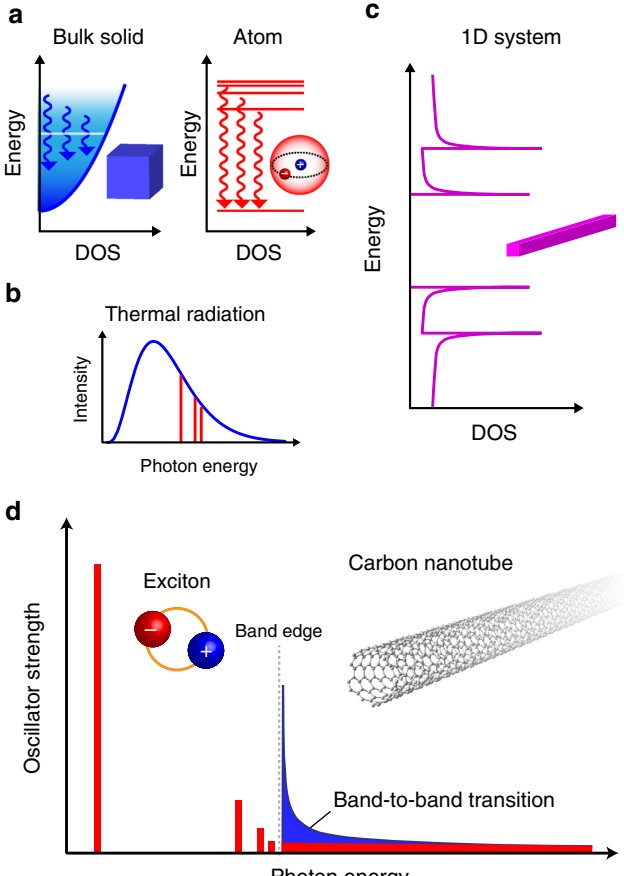

**Fig. 1** Optoelectronic characteristics of the one-dimensional system. **a** Electronic density of states (DOS) in a bulk solid and an atom. The white horizontal line and the vertical arrows indicate the Fermi energy and radiative transitions, respectively. The blue shading indicates the thermal distribution of electrons. **b** Typical thermal radiation characteristics of a bulk solid (blue) and an atom (red). **c** Electronic DOS in an one-dimensional (1D) semiconductor. **d** Schematic of the oscillator strength distribution of 1D systems in excitonic picture (red) or single-particle (band-to-band) picture (blue). The band edges are drawn aligned schematically for comparison. The inset shows a single-walled carbon nanotube as a representative quasi-1D system

## Results

**Laser heating and temperature evaluation of nanotubes.** To provide an ideal experimental system for observing intrinsic 1D high-temperature radiation, structure-defined, isolated, and pristine nanotubes were suspended in vacuum and laser-heated (Fig. 2a; see Methods for the detailed description). Continuous-wave (CW) laser irradiation provides non-contact local heating while retaining the neutral charge balance of the nanotubes

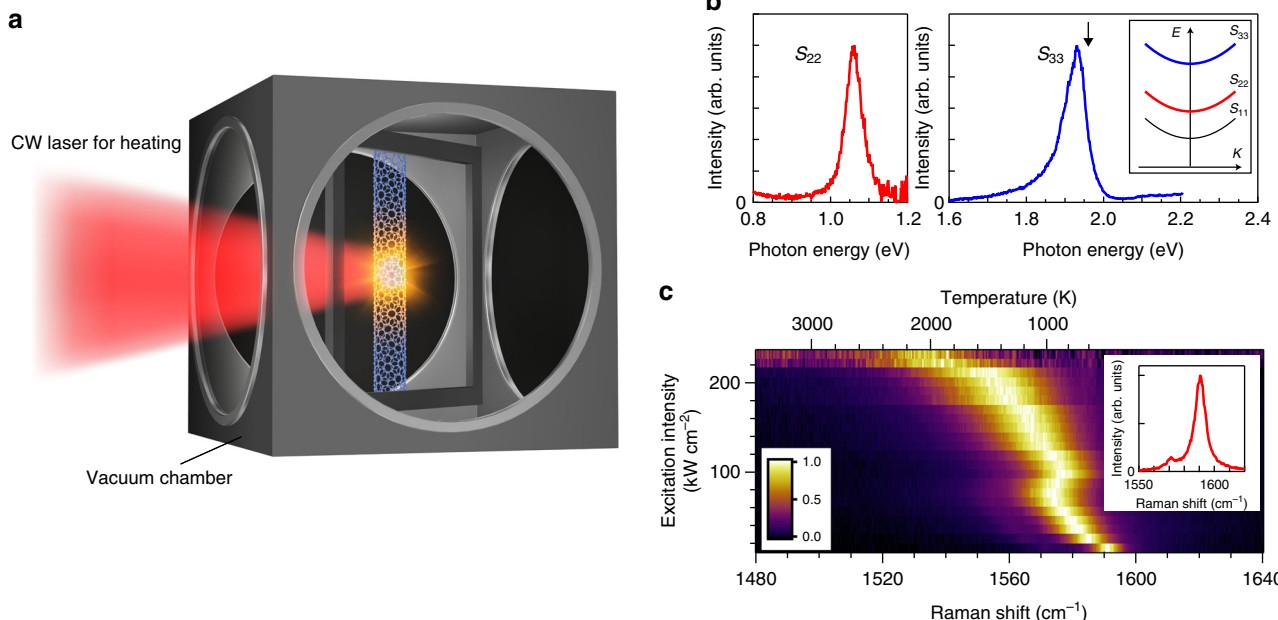

**Fig. 2** Laser heating and temperature evaluation of nanotubes. **a** Schematic of the experimental setup. **b** The near-infrared-to-visible range excitonic Rayleigh scattering spectra of the (18,8) semiconducting single-walled carbon nanotube. The black arrow indicates the photon energy of the helium–neon continuous-wave (CW) laser. The inset shows the exciton sub-band structures of the nanotubes ($S_{11}$, $S_{22}$, $S_{33}$). **c**, Two-dimensional plot of the excitation intensity dependence of the $G$-mode features of the Raman spectra. The top axis indicates the temperature estimated from the peak position of the $G$-mode. The inset shows the $G$-mode feature at 400 K

throughout the measurements. By suspending an individual nanotube over an open slit under vacuum, energy exchange with the surroundings is suppressed as far as possible. This study focuses on thermal spectra in the energy range around the higher-order 1D sub-bands ($S_{22}$ and $S_{33}$); this allows high-temperature radiation to be clearly distinguished from photoluminescence (PL), which predominantly occurs from the lowest energy sub-band ($S_{11}$; inset of Fig. 2b). An (18,8) semiconducting nanotube was prepared as a prototypical 1D semiconductor, together with a (30,12) metallic nanotube for comparison (see Methods). The $S_{22}$ exciton of the (18,8) nanotube was observed at 1.06 eV at room temperature (Fig. 2b)[24]. To achieve efficient laser heating, a helium–neon CW laser (1.959 eV), with resonance at the characteristic $S_{33}$ absorption at 1.93 eV, was used. The CW laser was focused on the nanotube using an objective lens, which enables sensitive detection through dark-field measurement. In addition, the CW laser irradiation enables temperature measurement using Raman spectroscopy of the in-plane carbon stretching mode ($G$-mode; inset of Fig. 2c)[25]. A gradual redshift of the $G$-mode frequency was observed as the excitation intensity increased (Fig. 2c), and the nanotube temperatures were estimated from the peak position according to the literature[25] (see Methods; the corresponding temperatures are indicated by the top axis).

**Thermal radiation of intrinsic semiconducting nanotubes.** Figure 3a shows a near-infrared-to-visible range optical image of the (18,8) nanotube above 2000 K. This light emission is linearly polarized along the nanotube axis (Fig. 3b), indicating that the 1D nature of the electronic states is stable even at 2000 K. Fig. 3c shows the light emission spectrum of the (18,8) nanotube at 1470 K. An intriguing feature is the much narrower

spectral linewidth (FWHM of ~170 meV) compared to that of the ideal blackbody radiation following Planck's law[1] at the same temperature (inset of Fig. 3c). It is initially confirmed that this light emission is thermally driven. The superlinear increase in the integrated emission intensity with the CW laser intensity ($I_{ex}$; inset of Fig. 3d) excludes the possibility that this emission originates from a normal PL process (see Methods for further confirmation). Replotting the emission intensities as a function of the inverse temperature clearly indicates a simple exponential behavior (Fig. 3d). This suggests a Boltzmann distribution proportional to $\exp(-E/k_B T)$, which describes the thermal population at a given energy ($E$); $k_B$ and $T$ are the Boltzmann constant and temperature, respectively. By fitting the experimental results to the Boltzmann distribution, it is found that $E = 0.80 \pm 0.02$ eV, which closely corresponds to the lower energy onset of the emission peak. This confirms that the light emission observed from the nanotube is thermally driven.

**Comparison between semiconducting and metallic nanotubes.** The high-temperature radiation peak energy probably originates from $S_{22}$ exciton resonance (see Figs. 2b, 3c). However, the possibility[19,20] of free electron–hole recombination at sharp van Hove singularities[26] in the 1D density of states (DOS) has not been excluded. To differentiate between these two possibilities, high-temperature radiation spectra from a metallic (30,12) nanotube were examined, since the 1D free electron–hole picture can be applied to hot (>1000 K) metallic nanotubes because of their small exciton binding energy (50 meV/$k_B$ ≈ 580 K)[27,28]. Fig. 4a, b show high-temperature radiation spectra of the semiconducting and metallic nanotubes at around 1400 K. Although the semiconducting nanotube exhibits a relatively symmetric spectral shape with a small

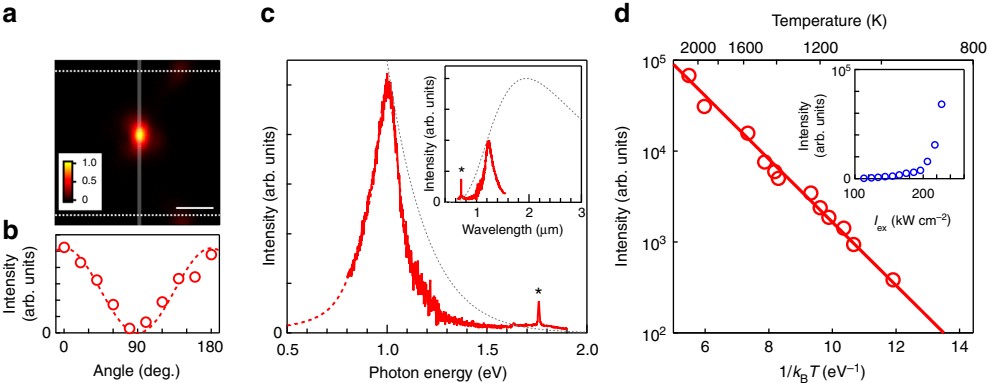

**Fig. 3** Thermally driven radiation of intrinsic semiconducting nanotubes. **a** Optical image of the light emission from the nanotubes at 2100 K. The gray vertical and white horizontal dotted lines indicate the positions of the nanotube and slit edges, respectively. The scale bar is 5 μm. **b** The polarization dependence of the emission intensities of the nanotube at 1470 K. The polarization is parallel to the nanotube axis at 0°. The dotted curve is a $\cos^2$ fit. **c** The light emission spectrum of the (18,8) nanotube at 1470 K. The red dotted curve is a guide to the eye (see Spectral analyses in Methods). In the inset, the same spectrum is replotted as a function of wavelength. The gray dotted curves indicate blackbody radiation at 1470 K. The peak intensity of the nanotube's emission is normalized to the blackbody radiation curve. The asterisks indicate the $G$-mode. **d** Linear-log plot of the emission intensities integrated over 0.8–1.2 eV as a function of the inverse temperature ($1/k_{\mathrm{B}}T$). The solid line is the fitting result based on the Boltzmann statistics described as $\exp(-E/k_{\mathrm{B}}T)$, with $E = 0.80$ eV. The inset shows the dependence of the emission intensities on the continuous-wave laser intensity ($I_{\mathrm{ex}}$)

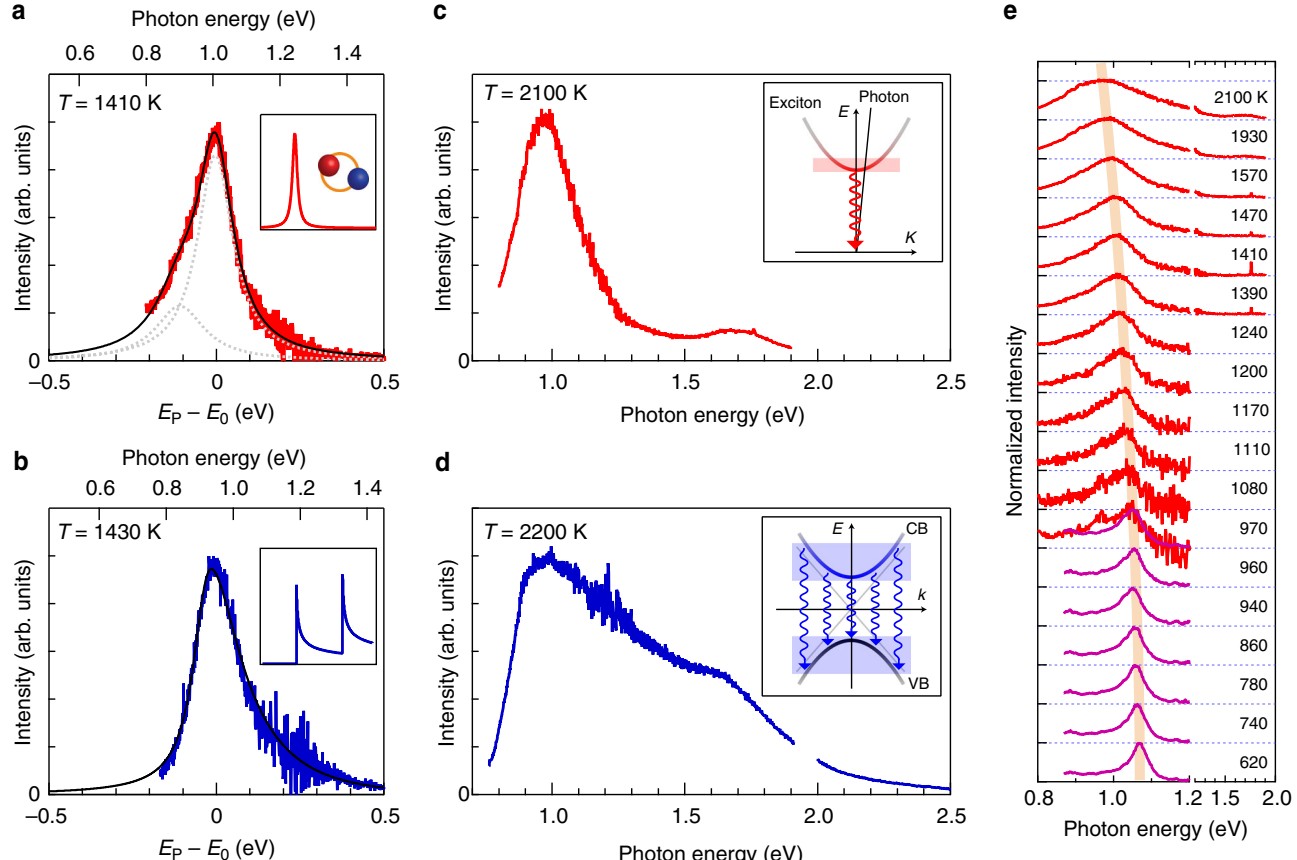

**Fig. 4** Emission spectra of thermally excited nanotubes. **a, b** High-temperature radiation spectra of the (18,8) semiconducting (**a**) and (30,12) metallic (**b**) nanotubes around 1400 K, respectively. The bottom axes show the differences from the peak photon energy ($E_{\mathrm{P}} - E_0$). The black solid curves and the gray dotted curves show the calculated results (see Methods for detailed spectral analyses). The inset shows the calculation models based on exciton (**a**) and free electron–holes in one-dimensional joint density of states (**b**). **c, d** Radiation spectra of the semiconducting (**c**) and metallic (**d**) nanotubes around 2100 K. The inset shows the dispersion relations of excitons and photons in the exciton picture (**c**) and those of the conduction band (CB) and valence band (VB) in the free electron–hole picture (**d**). The shaded regions indicate the energy–momentum states of excitons, electrons, and holes that can be involved in the radiative transitions (vertical arrows). **e** The high-temperature radiation (red curve) and Rayleigh scattering (purple curve) spectra of the (18,8) semiconducting nanotube at various temperatures. The spectra are vertically offset for clarity. Rayleigh scattering spectra could be measured up to ~970 K, and the high-temperature radiation spectra could be observed above ~970 K. In the high-temperature radiation region (>~1000 K), Rayleigh spectra could hardly be measured. The orange curve is a guide to the eye

shoulder at lower photon energies, the spectrum of the metallic nanotube has a rather long tail to the higher photon energy side (see Methods for detailed spectral analyses). This qualitative difference is more prominent at high temperatures above 2000 K (Fig. 4c, d); the metallic nanotube shows a much broader spectrum, which is indicative of free electron–hole recombination reflecting asymmetric 1D joint DOS with van Hove singularities (inset of Fig. 4b). The striking contrast between the spectral features indicates that their emission peaks have different origins. The ultra-narrow-band emission observed only from the semiconducting nanotube indicates the distinctive excitonic nature of the light emission (inset of Fig. 4a).

**Evidence for thermal generation of excitons**. The above attribution is also strongly supported by comparing the high-temperature radiation spectra with an optical susceptibility simultaneously examined using Rayleigh scattering spectroscopy (see Methods for the experimental details). As shown in Fig. 4e, the peak energies of the $S_{22}$ excitonic Rayleigh scattering and the high-temperature radiation nearly coincided at ~970 K, and the peak energy changes continuously from 620 to 2100 K. If the exciton is dissociated into a free electron–hole pair, a discontinuous increase in the peak energies due to their binding energies (at least >300 meV)[29] is expected during the heating process. The continuous peak shift confirms the stability of the 1D excitons even at very high temperatures above 1500 K. To our best knowledge, these experimental results represent the first conclusive evidence of the thermal generation of excitons in an intrinsic semiconductor, that is, the generation of a neutral exotic atom from a thermal bath. This may appear counterintuitive from the viewpoint of conventional semiconductor physics because, in conventional compound semiconductors such as GaAs[30], excitons are observed only at cryogenic temperatures under photoexcitation. However, the relatively small bandgap, the very large exciton binding energy, and the outstanding thermal stability of semiconducting carbon nanotubes enable the thermal generation of excitons.

## Discussion

The thermal exciton radiation of the intrinsic semiconducting nanotubes results in a very narrow spectral linewidth (Fig. 3c). The FWHM is about 170 meV around 1500 K, which is much narrower than that previously observed in Joule-heated and doped nanotubes at a similar temperature (about 350–400 meV)[19,20]. This suggests that the electronic correlation effects were less altered in our measurements, which led to successful observation of the inherent properties of the intrinsic 1D semiconductor. In contrast to free carrier recombination (inset of Fig. 4d), only excitons with near-zero momentum can recombine radiatively because of the energy–momentum conservation requirement for exciton photoemission (inset of Fig. 4c), which enables narrow-band light emission. The strongly suppressed high-energy radiation compared to the blackbody radiation curve at the same temperature (Fig. 3c) is remarkable; this is difficult to achieve in the near-infrared-to-visible range even using previous metal or semiconductor nanophotonic thermal emitters[4–6] in contrast to the successful control of the thermal emission in the mid-infrared (~10 μm) wavelength region (Q factor of over 100) using quantum wells and photonic crystals[31,32]. To our best knowledge, this peaked thermal exciton radiation has the narrowest FWHM among reported solid state thermal emitters operable in the near-infrared-to-visible range around 1500 K[3–7]. The structure-

dependent exciton resonance energies allow the nanotubes to potentially serve as ultra-narrow-band thermal emitters at various wavelengths. Furthermore, we also confirmed that intrinsic semiconducting nanotubes persistently exhibit the narrow-band thermal exciton radiation for more than 10 h even at ~2000 K; this demonstrates remarkable thermal stability of carbon nanotubes under the high temperature conditions (Supplementary Fig. 1).

Finally, we comment on the potential significance of our findings with respect to a specific optothermal application known as thermophotovoltaics[2,3]. If the thermal energy stored in a solid can be efficiently converted to monochromatic near-infrared light, it can be used as an input to a photovoltaic cell to generate electricity (Supplementary Fig. 2a). The narrower the spectral linewidth at the near-bandgap energy of the photovoltaic cell, the higher the energy conversion efficiency because of the photon energy-dependent conversion efficiency of standard photovoltaic cells (Supplementary Fig. 2b). It would be reasonable to use the ultra-narrow-band thermal exciton radiation of semiconducting nanotubes at >1000 K for this specific application. Thus, our findings may offer a new design principle (see Supplementary Note 1 for a detailed discussion) for a narrow-band thermal emitter that enables efficient thermophotovoltaic energy harvesting, as well as for initiating a new field of high-temperature exciton science and engineering.

## Methods

**Synthesis and characterization of nanotubes**. Nanotubes were grown and suspended over open slits of 20–30 μm width, which were cut into substrates, by an ambient chemical vapor deposition method using a modified fast-heating process[33]. The chiral indices were determined using an empirical formula describing the relation between the structures and the optical exciton resonance[24]. The optical exciton resonances were probed by means of elastic scattering (Rayleigh) spectroscopy[34], in which scattering is enhanced at the exciton resonances. The Rayleigh scattering cross-section is proportional to $\omega^3|\chi(\omega)|^2$, where $\omega$ and $\chi(\omega)$ are the optical frequency and the susceptibility, respectively[35]. All of the Rayleigh spectra are corrected for the $\omega^3$ scattering efficiency factor to show the optical susceptibility. Broadband light from a supercontinuum source (Fianium, WL-SC-400-PP-4) was focused on an individual nanotube through an objective lens with a numerical aperture of 0.42 (Supplementary Fig. 3). The integrated power was about 2 mW (0.56–2.8 eV). The scattered light collected by another objective lens with a numerical aperture of 0.42 was detected with a monochromator attached to a thermoelectrically cooled charge-coupled device camera (Princeton Instruments, ProEM) or one attached to a thermoelectrically cooled indium–gallium–arsenide camera (Princeton Instruments, NIRvana). By comparison with the empirical formula[24], peaks at 1.06 and 1.93 eV shown in Fig. 2b are assigned to the second ($S_{22}$) and third ($S_{33}$) sub-band excitons of the (18,8) semiconducting nanotube with a diameter of 1.81 nm. This assignment was validated by measuring the radial breathing modes of the Raman spectra, the frequency of which is inversely proportional to the nanotube diameter (Supplementary Fig. 4a)[36]. The structure of the (30,12) metallic nanotube was determined in the same way (Supplementary Figs. 4b, c), and the resonance peaks of the Rayleigh spectrum in the near-infrared and visible regions are assigned to the first and second sub-band ($M_{11}$ and $M_{22}$) exciton states, respectively. The structure of the (15,11) semiconducting nanotube used for evaluating the durability (Supplementary Fig. 1) was uniquely determined from the Rayleigh spectrum (Supplementary Fig. 4d); the three resonance peaks are assigned to the second, third, and fourth ($S_{22}$, $S_{33}$, and $S_{44}$) sub-band exciton states of the (15,11) nanotube.

**Estimation of the temperature using Raman spectra**. It was reported that the Raman G-mode frequency of nanotubes has a universal temperature dependence, which is independent of sample type or laser excitation wavelength[25]. An empirical law for the temperature (T)-dependent G-mode frequency, $\omega(T)$, was proposed[25] as

$$\omega(T) = \omega_0 - \frac{A}{\exp(B\hbar\omega_0/k_BT) - 1}, \quad (1)$$

where $\omega_0$, $\hbar$, and $k_B$ are the zero-temperature G-mode frequency, the reduced Planck constant, and the Boltzmann constant, respectively. Parameters $\omega_0$, $A$, and $B$ are given as $\omega_0 = 1594\ \mathrm{cm}^{-1}$, $A = 38.4\ \mathrm{cm}^{-1}$, and $B = 0.438$ according to ref. [25].

**Exclusion of a non-thermal PL process**. A nanotube in a nitrogen atmosphere was subjected to CW laser irradiation; this enabled photoexcitation of the nanotube while avoiding a rise in its temperature, as it was subjected to convection cooling. Negligible light emission was observed for the corresponding nanotube under the CW laser intensity at which the nanotube exhibited photoemission in a vacuum condition. This confirms that the observed light emission does not originate from non-thermal PL processes.

**Spectral analyses**. The radiation spectrum of the semiconducting nanotube is analyzed using the exciton picture (Figs. 3c, 4a). It was reported that the phonon sideband emission lies ~100 meV below the exciton state[37]. The emission spectrum is fitted to the sum of the two Lorentzian functions by considering the exciton and phonon sideband emissions (indicated by the gray dotted curves in Fig. 4a). The guide to the eye (red dotted curve) in Fig. 3c is drawn using the same method. The emission spectrum of the metallic nanotube is analyzed using the model by considering free electron–hole recombination in the 1D DOS with van Hove singularities[19,38]. The emission spectrum of $I(E)$ in the observed energy range was modeled as $I(E) = A_R(E) \, D_J(E) \, f_{FD}[E_c(k)]\{1 - f_{FD}[E_v(k)]\}$. Here, $E_{c(v)}(k)$ is the dispersion relation for the conduction (valence) band, and the emission energy $E$ is given by $E = E_c(k) - E_v(k)$. Because of the nearly symmetric conduction and valence bands of nanotubes, it was assumed that $E_c(k) = -E_v(k)$. $f_{FD}(E)$ is the Fermi–Dirac distribution. $D_J(E)$ is the joint DOS for the 1D system, and the splitting of the first pair of massive 1D metallic sub-bands ($M_{11L}$ and $M_{11H}$) due to the trigonal warping effect is considered for the (30,12) metallic nanotube (see Supplementary Fig. 4b). $A_R(E)$ is the radiative recombination rate ($\propto \sim E$)[19]. The result is calculated by convolving $I(E)$ with a Lorentzian function, and this satisfactorily reproduces the characteristic long tail to higher photon energies (indicated by the black curve in Fig. 4b).

**Temperature dependence of the optical susceptibility**. Variations in the optical susceptibility were examined over a wide temperature range by measuring the light emission ($\propto$ an imaginary part of $\chi(\omega)$) and light (Rayleigh) scattering ($\propto |\chi(\omega)|^2$) spectra (Supplementary Fig. 5). At temperatures where no light emission was detectable, the optical susceptibilities were probed by Rayleigh spectroscopy. Both the broadband light (for Rayleigh spectroscopy) and CW laser (for heating and Raman spectroscopy) were concentrated on the same region of the nanotube at a temperature below 970 K. When the temperature reached 970 K, thermally driven radiation was emitted. At 970 K, both measurements of the thermally driven radiation and Rayleigh scattering spectra were possible because the intensity of the thermally driven radiation was observed to be still weak enough to measure the Rayleigh scattering spectra. Above 970 K, the thermally driven emission intensity was greater than that of the Rayleigh scattering signal, and only this radiation was observed under CW laser irradiation.

**Data availability**. The data that support the findings of this study are available from the corresponding authors upon reasonable request.

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

## Acknowledgements

This work was supported by the ERATO program from JST (K.I.) (grant number: JPMJER1302). Part of this work was supported by JSPS KAKENHI Grant Number JP24681031 (Y.M.). We thank Keisuke Matsui (Nagoya University) for assistance of synthesis and finding nanotubes, and Takahiro Yamamoto (Tokyo University of Science) and Satoru Konabe (Tokyo University of Science) for the helpful discussions.

## Author contributions

T.N. and Y.M. conceived the concept and K.I. directed the project. A.T. synthesized the nanotubes. T.N. arranged and carried out all optical experiments. T.N. and Y.M. considered the mechanism. All the authors contributed to writing the manuscript.

## Additional information

**Competing interests:** The authors declare no competing interests.

