## [Peer Review File · Nature Communications]

Reviewer #1 (Remarks to the Author):

I thank the authors for highlighting the difference of engineering both the electronic and the photonic DOS compared to the previous work i mentioned. As a result, the authors claim that this process "inherently enhances the radiative transition rate of excitons much more than that of the free electron-hole recombination, and leads to efficient and ultra -narrow-band thermal radiation."

1. While this is partially true, it is not novel. The idea of engineering both electronic and photonic DOS to achieve narrowband thermal emission was already realised by Noda's group (Nat Phot 2012 and APL 2013). In fact, the APL 2013 paper reports a Q-factor of over 100, which is much narrower than what is reported here. The authors should refer to this work and also quote Q-factors.

Secondly, all of the systems discussed so far essentially act as filters for the Planck curve, and whether only the photonic DOS or both the photonic and electronic DOS are manipulated, the result is the same, namely that the Planck curve cannot be exceeded. Therefore, talking about "efficient" radiation and "unprecedentedly narrow-band" emission, as done by the authors here, is misleading and should be avoided. Nevertheless, the fact that such narrowband emission is observed at much higher temperature than by other authors is quite remarkable and novel.

2. The authors show data for temperatures as high as 2100K. How long can the CNTs be maintained at that temperature ? Can the authors show date of emission vs time ?

3. Finally, I appreciated the discussion about practicality, i.e. the idea of packing CNTs with hBN spacers, which I agree will require further work.

Reviewer #2 (Remarks to the Author):

The authors answers correctly for the previous reviewers' comments for Nature Physics. The information of narrow spectra of PL for semiconducting at 2000K should be a new information in nanotube optics.

Thus as one of the previous referee, I should not give the comment further.

After reading the manuscript again, I am wondering how the oscillator-strength sum rule (or f-sum rule) for the exciton works in this phenomena. May I understand that the PL spectra of semiconducting CNT is given by the product of (1) square of oscillator-strength (exciton + single particle excitation) as a function of energy, (2) joint density of states, and (3) Fermi distribution function? As for the metallic nanotubes, the authors estimate the intensity as $A_R D_J f_c(1-f_v)$.

Response Letter

We are sending our revised manuscript entitled “Ultra-narrow-band near-infrared thermal exciton radiation in intrinsic one-dimensional semiconductors” by Taishi Nishihara, Akira Takakura, Yuhei Miyauchi, and Kenichiro Itami for publication in *Nature Communications*. To enhance the clarity of our claim of the manuscript, we modified the title (we added a word "near-infrared"). We have carefully revised the manuscript in view of the constructive and helpful reviewers' comments. Below we present the Summary of Changes in the manuscript.

Summary of Changes

- We added a word "near-infrared" in the title according to Q1-1).
- In abstract (line 6), we removed a word according to Q1-1).
- On page 4 (3rd line in 2nd paragraph), we removed a word according to Q1-1).
- On page 8 (lines 7–11 in 1st paragraph), we added words and a sentence according to Q1-1).
- On page 8 (line 14 in 1st paragraph), we added a sentence according to Q1-2).
- On page 9 (line 3 in 1st paragraph), we removed a word according to Q1-1).
- On page 10 (line 2 in 1st paragraph), we added sentences according to Q1-2).
- We added the references (No. 31 and 32) according to Q1-1).
- We added Supplementary Fig. S1 and Fig. S4d according to Q1-2).

Responses to the Reviewers' comments and suggestions

We are grateful for the reviewers' important comments and suggestions for improving the manuscript. Using the reviewers' insightful comments, we have revised the manuscript. In the following, point-by-point responses to the reviewers' comments and the revisions included in the revised manuscript are presented.

For Reviewer #1:

Remarks to the Author

I thank the authors for highlighting the difference of engineering both the electronic and the photonic DOS compared to the previous work I mentioned. As a result, the authors claim that this process "inherently enhances the radiative transition rate of excitons much more than that of the free electron-hole recombination, and leads to efficient and ultra-narrow-band thermal radiation."

Q1-1) *While this is partially true, it is not novel. The idea of engineering both electronic and photonic DOS to achieve narrowband thermal emission was already realized by Noda's group (Nat Phot 2012 and APL 2013). In fact, the APL 2013 paper reports a Q-factor of over 100, which is much narrower than what is reported here. The authors should refer to this work and also quote Q-factors.*

We thank the reviewer for finding the missing of the important references. Although we are convinced that experimental confirmation of the *thermal exciton generation* phenomena and the use of this mechanism for realizing *near-infrared* narrow-band thermal emission are novel, the concept of engineering both the electronic and the photonic DOS has already been reported in the literatures as the reviewer suggested. We added the comment on the successful spectral controls of the *mid-infrared* thermal emission, realized by manipulating photonic DOS using photonic crystals and quantum wells. According to this comment, we added words at **page 8, line 7 in 1st paragraph** as "near-infrared-to-visible range even using", "metal or semiconductor nanophotonic", and "solid state" (at **page 8, line 11 in 1st paragraph**), a sentence at **page 8, line 8 in 1st paragraph** as "in contrast to the successful control of the thermal emission in the mid-infrared (~10 μm) wavelength region (Q factor of over 100) using quantum wells and photonic crystals^{31,32}", and the references (No. 31 and 32).

Secondly, all of the systems discussed so far essentially act as filters for the Planck curve, and whether only the photonic DOS or both the photonic and electronic DOS are manipulated, the result is the same, namely that the Planck curve cannot be exceeded. Therefore, talking about "efficient" radiation and "unprecedentedly narrow-band" emission, as done by the authors here, is misleading

and should be avoided. Nevertheless, the fact that such narrowband emission is observed at much higher temperature than by other authors is quite remarkable and novel.

We thank the reviewer for finding our observation of the narrowband exciton emission in the near-infrared range at very high temperature of quite remarkable and novel. To distinguish our findings to previous results clearly, we added a word "near-infrared" in the title. We agree with the reviewer's comment that "the Planck curve cannot be exceeded" by the thermal emission process. According to this comment (of which concerns are shared with the editor), we removed "unprecedentedly" (at page 2, line 6 in abstract and at page 4, 3rd line in 2nd paragraph), and "efficient" (at page 9, line 3 in 1st paragraph) to avoid misleading.

Q1-2) *The authors show data for temperatures as high as 2100K. How long can the CNTs be maintained at that temperature? Can the authors show date of emission vs time?*

To clarify this issue, we measured the temporal evolution of the thermal exciton radiation. Since it is difficult to prepare a semiconducting nanotube with the identical (n, m) shown in the manuscript using our growth method (because the structures of the nanotubes obtained in each growth experiment are nearly random), we used the (15,11) semiconducting nanotube that showed thermal exciton radiation with photon energies similar to those of (18,8) nanotube (Fig. S4d). During the experiment over 10 hours, the spectral shape (Fig. S1) and intensities (inset of Fig. S1) of the thermal exciton radiation were almost unchanged (and we did not observe breakdown of the nanotube during the 10 hours experiment), which confirms remarkable thermal stability of carbon nanotubes. According to this comment, we added a sentence at page 8, line 14 in 1st paragraph as "Furthermore, we also confirmed that intrinsic semiconducting nanotubes persistently exhibited the narrow-band thermal exciton radiation for more than 10 hours even at ~2,000 K; this demonstrates remarkable thermal stability of carbon nanotubes under the high temperature conditions (Supplementary Fig. S1).", and Supplementary Fig. S1 at page 2 in SUPPLEMENTARY INFORMATION. We also added the information of the (15,11) nanotube at page 10 line 2 in 1st paragraph as "The structure of the (15,11) semiconducting nanotube used for evaluating the durability (Supplementary Fig. S1) was uniquely determined from the Rayleigh spectrum (Supplementary Fig. S4d); the three resonance peaks were assigned to the second, third, and forth (S_{22} , S_{33} , and S_{44}) sub-band exciton states of the (15,11) nanotube.", and Supplementary Fig. S4d at page 3 in SUPPLEMENTARY INFORMATION.

Q1-3) *Finally, I appreciated the discussion about practicality, i.e. the idea of packing CNTs with hBN pacers, which I agree will require further work.*

Again, we thank the reviewer for finding the proposed thermophotovoltaic application of interest. As a next step, we are going to realize such device applications. We hope that we will report them in future communications.

For Reviewer #2:

Remarks to the Author:

The authors answers correctly for the previous reviewers' comments for Nature Physics. The information of narrow spectra of PL for semiconducting at 2000K should be a new information in nanotube optics.

Thus as one of the previous referee, I should not give the comment further.

After reading the manuscript again, I am wondering how the oscillator-strength sum rule (or f-sum rule) for the exciton works in this phenomenon. May I understand that the PL spectra of semiconducting CNT is given by the product of (1) square of oscillator-strength (exciton + single particle excitation) as a function of energy, (2) joint density of states, and (3) Fermi distribution function? As for the metallic nanotubes, the authors estimate the intensity as $A_R D_J f_c(1-f_v)$.

We thank the reviewer for reviewing our manuscript again. In the case of thermal excitonic radiation, we cannot simply use $A_R D_J f_c(1-f_v)$ because of the reason as follows: In semiconducting carbon nanotubes, in contrast to common semiconductors, oscillator strengths for the band-to-band transitions in the low energy subbands are almost missing, because of the substantial concentration of the oscillator strength to the 1D exciton state. This is a unique characteristic of 1D semiconducting systems [Ref. 14 in the main text], and one of the major consequences of the f-sum rule. In such a situation, we can neglect the band-to-band transitions and only an exciton state should be taken into account. The light emission process of this case is quite similar to that of exciton luminescence, in which the exciton radiative decay rate is proportional to the oscillator strength (\propto square of the transition dipole moment) and the square of the corresponding photon energy (\propto photonic DOS). Thus, the major difference between normal photoluminescence and thermal exciton radiation processes can be understood as the creation mechanisms of the excitons (by photogeneration (\propto incident light intensity) or by thermal generation (\propto Boltzmann distribution)). As for the exciton's radiative decay, in addition to the photonic

DOS, one also has to consider a portion of the excitons that can contribute to the radiative transition with satisfying the exciton-photon momentum conservation, among thermalized ones in the 1D exciton band (Inset of Fig. 3c). This portion depends on the exciton linewidth. (for more details, please see [J. Feldmann, *et al.*, Phys. Rev. Lett. 59, 2337 (1987)] and [Y. Miyauchi, *et al.*, Phys. Rev. B 80, 081410(R) (2009)]). With regard to the metallic ones, in contrast, the use of $A_{RD} J_{fc}(1-fv)$ is justified because the exciton binding energy for the metallic nanotubes is much smaller than the semiconducting counterpart [Ref. 27], and one can attribute the thermal radiation to the band-to-band transitions in massive 1D subbands of the metallic nanotubes, as has been discussed in the literature [Refs. 19, 20].

Reviewer #1 (Remarks to the Author):

I thank the authors for implementing my suggestions and am happy for the paper to proceed to publication.

Reviewer #2 (Remarks to the Author):

The authors answered the further questions by Referee. Thus Referee agrees that the paper is published in Nat. Commun.

Responses to the Reviewers' comments

Reviewer #1 (Remarks to the Author):

I thank the authors for implementing my suggestions and am happy for the paper to proceed to publication.

Reviewer #2 (Remarks to the Author):

The authors answered the further questions by Referee. Thus Referee agrees that the paper is published in Nat. Commun.

Again, we thank the reviewers for their important comments and suggestions for improving the manuscript.